# Coexisting Biopsy-Diagnosed Dementia and Glioblastoma

**DOI:** 10.3390/brainsci14020143

**Published:** 2024-01-30

**Authors:** Kaleigh Fetcko-Fayad, Kristen Batich, Zachary J. Reitman, Kyle M. Walsh, Gregory Chamberlin, Vanessa Smith, Karra Jones, Thomas Cummings, Katherine B. Peters

**Affiliations:** 1Department of Neurosurgery, Neuro-Oncology, Duke University Medical Center, Durham, NC 27710, USA; kristen.batich@duke.edu (K.B.); katherine.peters@duke.edu (K.B.P.); 2Department of Radiation Oncology, Duke University Medical Center, Durham, NC 27710, USA; zjr@duke.edu; 3Department of Neurosurgery, Neuro-Epidemiology, Duke University Medical Center, Durham, NC 27710, USA; kyle.walsh@duke.edu; 4Department of Pathology, Clinical Pathology Services, Duke University Medical Center, Durham, NC 27710, USA; gregory.chamberlin@duke.edu (G.C.); vanessa.l.smith@duke.edu (V.S.); karra.jones@duke.edu (K.J.); thomas.cummings@duke.edu (T.C.)

**Keywords:** glioblastoma, dementia, Alzheimer’s disease, cerebral amyloid angiopathy

## Abstract

Both glioblastoma (GBM) and dementia are devastating diseases with limited treatments that are usually not curative. Having clinically diagnosed dementia with an associated biopsy-proven etiology and a coexisting GBM diagnosis is a rare occurrence. The relationship between the development of neurodegenerative dementia and GBM is unclear, as there are conflicting reports in the literature. We present two cases of simultaneous biopsy-proven dementia, one with Alzheimer’s disease (AD) and GBM, and one with cerebral amyloid angiopathy (CAA) and GBM. We discuss how these diseases may be associated. Whether one pathologic process begins first or develops concurrently is unknown, but certain molecular pathways of dementia and GBM appear directly related while others inversely related. Further investigations of these close molecular relationships between dementia and GBM could lead to development of improved diagnostic tools and therapeutic interventions for both diseases.

## 1. Introduction

Both glioblastoma (GBM) and dementia are devastating diseases with limited treatments that are usually not curative [1]. Having clinically diagnosed dementia with an associated biopsy-proven etiology and a coexisting GBM diagnosis is a rare occurrence [2,3]. The relationship between the development of neurodegenerative dementia and GBM is unclear [3]. Some studies have observed an inverse correlation between cancer and neurodegenerative disease incidence, with cancer peaking around age 60 years, but beyond that, neurodegenerative diseases increase in the elderly [4]. Other studies have noted the peak incidence of dementia around age 65–95 years, while GBM usually appears around age 65–85 years [5]. Lehrer postulated a potential mechanistic relationship between AD and GBM given that AD prevalence and GBM incidence paralleled in some US states [6]. Certain similarities between dementia and GBM suggest a direct association, including neurodegeneration-associated protein deposits, inflammation, and intestinal microbiota alterations [1,3].

AD is the most common neurodegenerative disease [3]. Though the exact mechanism of injury is unknown, accumulation of abnormally folded amyloid-β (Aβ) plaques and hyper-phosphorylated tau neurofibrillary tangles with the presence of cerebral amyloid angiopathy (CAA) are present [6,7]. Age-related cognitive decline involves both neurodegenerative and cerebrovascular processes; after AD, vascular pathologies—such as ischemic infarctions, atherosclerosis, arteriolosclerosis, and CAA—were among prominent findings in aging brains [7]. Although CAA can often be seen in AD neuropathology, these are two separate diagnoses with different amyloid characteristics. CAA occurs with cerebrovascular Aβ deposition, while AD consists of parenchymal Aβ deposition that primarily constitutes neuritic plaques; both processes have reduced perivascular clearance of Aβ but otherwise differ in regards to Aβ subtype composition, pattern of deposition, genetic links, and co-deposited components [7].

We have included two case descriptions of simultaneous biopsy-proven dementia (one with AD, one with CAA) and GBM pathologies. We further discuss how these diseases may be associated, which could potentially affect the development of further prognostic, diagnostic, or therapeutic tools for these possibly related diseases.

## 2. Case 1

A 68-year-old male presented with gradually worsening symptoms of severe fatigue, inattentiveness, urinary incontinence, and memory impairments over seven months. Cognitive symptoms continued to escalate prompting an evaluation in the emergency department. Magnetic resonance imaging (MRI) of his brain revealed an area of evolving parenchymal hemorrhage in the right insula (Figure 1a), with innumerable foci of hemosiderin in cerebral and cerebellar hemispheres (Figure 1b) and a T2/FLAIR hyperintense mass in the left insular region without contrast enhancement (Figure 1c,d).

Left insular mass biopsy revealed GBM CNS WHO grade 4 in the background of cerebral amyloid angiopathy (Figure 2a–c). Microscopic examination of biopsy tissue did not show any mitotic figures and lacked microvascular proliferation or necrosis; however, molecular studies resulted as isocitrate dehydrogenase (*IDH*)-wildtype, telomerase reverse transcriptase (*TERT*) promoter mutation (*C250T*), and chromosome 7 gain. DNA extraction was then performed (Mayo Clinic Laboratories, Rochester, MN, USA). For *IDH1*, *IDH2*, and *TERT* promoter mutation analysis, the extracted DNA was subjected to targeted next-generation sequencing (Mayo Clinic Laboratories, Rochester, MN, USA). Chromosomal microarray was performed using the Oncoscan platform (Applied Biosystems (Affymetrix), Waltham, MA, USA). It is key to point out that this was a molecular diagnosis of GBM based on CNS WHO 2021 criteria given *IDH*-wildtype and *TERT* promoter mutation despite the lack of contrast enhancement on his MRI; additional whole exome next-generation sequencing via Caris Life Sciences also revealed a likely pathogenic *PIK3CA* mutation (*c.1034A>T*).

He presented to our clinic experiencing more cognitive dysfunction, including an inability to work and drive. He received three-week concurrent hypofractionated radiation therapy (total dose of 40 Gy in 15 fractions) with oral temozolomide (TMZ) 75 mg/m^2^ daily, with which he was compliant. His formal diagnosis from the memory disorders clinic was suspected mixed dementia attributed to cerebral amyloid angiopathy as well as likely AD. He completed chemoradiation but continued to decline cognitively and functionally, so he ultimately transitioned to hospice care.

## 3. Case 2

A 69-year-old male presented with a steep decline in short-term memory and difficulty maintaining attention. The patient had a family history of dementia (father in late 50s and sister at age 60). Given his cognitive decline, his primary care physician initiated donepezil and memantine and ordered MRI brain, which showed diffuse atrophy with a non-contrast enhancing, T2/FLAIR hyperintense lesion in the left frontal lobe. This lesion persisted on a follow-up brain MRI one month later (Figure 3a,b). The patent underwent a biopsy of the left frontal lesion, which showed histopathologic evidence of an astrocytic glioma *IDH*-wildtype and *TERT* promoter mutation; a significant abundance of amyloid-β plaques and tau protein was also present within this biopsy specimen.

Six weeks later, he underwent maximal resection for optimal tumor cellular debulking to minimize neurologic deficits. This pathology similarly resulted as a diffuse astrocytic glioma, *IDH*-wildtype with high *TERT* promoter mutation (*C228T*) and Ki-67 staining of 10–20%, leading to an integrated diagnosis of GBM, CNS WHO grade 4 (Figure 2d). For IDH1, IDH2, and TERT promoter mutation analysis, the extracted DNA was subjected to PCR amplification followed by Sanger sequencing (Duke University Hospital, Durham, NC, USA). Chromosomal microarray was performed using the Oncoscan platform (Applied Biosystems (Affymetrix), Waltham, MA, USA). EGFR, CDKN2A, and PTEN FISH testing was performed using probes for 7p12, 9p21, and 10q23, respectively, as well as probes for the corresponding centromere (Abbott Molecular, Des Plaines, IL, USA). Further whole-genome next-generation sequencing via Caris Life Sciences revealed a pathogenic *PIK3CA* (*c.1636C>A*) mutation.

The resected specimen revealed cortical plaques with positivity for beta-amyloid precursor protein (APP) and positive tau staining in neurons and plaques (Figure 2e,f). Of note, this patient also was diagnosed with a molecular GBM but initially presented with a non-enhancing lesion on MRI. At his initial clinic visit, an additional Montreal Cognitive Assessment screen resulted in a 10/30 score, indicating severe executive and memory function deficits. An F-18 FDG brain PET scan demonstrated hypometabolism in the bilateral temporal lobes and, to a lesser extent, in the bilateral parietal lobes (Figure 3c,d). Formal neurocognitive testing revealed cognitive deficits in all modalities tested, but no significant behavioral, affective, or gait component was identified. The patient was ultimately diagnosed with Alzheimer’s dementia.

A repeat MRI brain about 8 weeks later revealed a new heterogeneously contrast-enhancing lesion in the ipsilateral cingulate gyrus with accompanying hyperintensity of T2/FLAIR signal. He was having increasing difficulty naming objects, performing familiar tasks, and discriminating previously familiar objects. Given the rapid development of this lesion and severe cognitive impairment, hypofractionated radiotherapy (total dose of 40 Gy in 15 fractions) with concurrent TMZ 75 mg/m^2^ po daily followed by adjuvant TMZ was recommended. The patient was compliant with the treatment regimen. However, after two cycles of adjuvant TMZ, MRI brain showed a slight decrease in size of the nodular-enhancing lesion in the left anterior medial frontal lobe with an increase in surrounding T2 signal with slight mass effect in the left frontal lobe and extending into the corpus callosum, concerning for early disease progression versus post-radiation treatment changes. The patient continued TMZ for another two 28-day cycles with a repeat MRI brain showing interval growth of the contrast-enhancing left medial frontal treated mass with associated hemorrhage and new contrast-enhancing extension along the corpus callosum and new small satellite lesions. After lengthy discussions with the patient and his spouse and in efforts to preserve his remaining quality of life, he transitioned to hospice care. He passed approximately 13 months after his initial biopsy.

## 4. Discussion

In cases with both dementia and GBM, it is unclear which disease began first or if they developed concurrently. Furthermore, how coexisting neurodegenerative and neoplastic processes interact and whether these interactions influence the clinical symptoms and disease course is unknown [5]. Though GBM induces an immunosuppressive microenvironment and the aging brain is considered more pro-inflammatory, both pathologic states contain reactive astrocytes and immunosuppressant microglial cells [8]. Disruption of astrogenic balance appears to be at play in both dementia and GBM, with cell cycle dysregulation leading to the development of malignant astrocytes, while astrocytic senescence and cell death without renewal results in neurodegeneration [4]. Depending on the type of localized inflammation or injury that occurs and which particular set of genes are affected, morphologic changes can occur, forming either brain cancer, neurodegenerative disease, or both [4].

In these two patients, a hypofractionated chemoradiation course was selected based on several clinical and radiobiological considerations. Hypofractionated courses of 40 Gy in 15 fractions may result in similar outcomes for elderly patients or those with lower performance status compared with standard-of-care courses such as 60 Gy in 30 fractions [9,10]. Using biologically effective dose modeling, we estimated the hypofractionated regimen would have a favorable effect on non-tumor brain tissue compared to a longer course, which may be important with comorbid CNS pathologies present [11]. We estimated normal brain to have an alpha/beta ratio of 2 (BED2), often used for slow-growing normal tissues. The biologically effective dose for BED2 for a hypofractionated regimen of 40 Gy in 15 fractions is 93.3 Gy^2^—lower than for a long-course regimen of 60 Gy in 30 fractions at 120 Gy^2^. The hypofractionated course facilitated feasible treatments for these patients with the T2-hyperintense expansile tumor area considered in gross tumor volume given the predominantly non-enhancing nature of the tumors [9,10]. Smaller clinical target volume expansions were used (5 mm) given concern for damage to normal brain structures [10].

A study examining genetic profiles found 40 genes expressed oppositely in cancer and neurodegenerative disease with specifically four transcription factors involved in regulating cellular proliferation, inflammation, and apoptosis: activator protein 1 (*AP-1*), nuclear factor of activated T cells (*NFAT*), CCAAT/enhancer-binding protein beta (*C/EBPβ*), and E2F transcription factor 1 (*E2F1*) [4]. Candido and colleagues discovered a strong negative association between miRNA levels that were deregulated in both AD and GBM, which participated in the regulation of genes expressed in glioma pathways (such as *MAPK1*, *IGF1R*, *PIK3*, *RAS*), as well as the expression of genes involved in AD pathology (such as *APP*, *GSK3B*, *NDUF*, and *LRP1*)—many of which were associated with both cellular and metabolic processes [1]. Liu also conducted a comprehensive bioinformatics study that identified significant transcriptional pathways inversely regulated in GBM and AD patients. Notable pathways upregulated in GBM and downregulated in AD included cAMP signaling, IL-3 signaling, IGF-1 signaling, axonal guidance signaling, ERK/MAPK signaling, and inhibition of angiogenesis by TSP1, while significant pathways downregulated in GBM and upregulated in AD consisted of interleukin signaling, PDGF signaling, PPAR signaling, TNFR1 signaling, NF-kB signaling, and angiopoietin signaling [12]. Via *in vitro* and *in vivo* mouse models, their results showed that elevated Aβ may lead to reduced ERK1/2 signaling in AD, which is intrinsically activated in GBM; additionally, Aβ appeared to inhibit GBM cell proliferation, invasion, and migration [12].

By evaluating The Cancer Genome Atlas, another study found 1643 somatic mutations of dementia-related genes, including *APP*, *PSEN1*, *PSEN2*, *MAPT*, *SCNA*, and *LRRK2*, in cancer patients, yet an extremely low distribution was noted in GBM patients specifically [13]. However, the American Tissue Culture Collection of gliomas and GBM noted that *APP*, in particular, was upregulated, and Aβ deposition was also found in glial tumors and neighboring blood vessels in glioma mouse models [14]. Similar to the neuropathologic nature of AD, GBM is also known to interact with the surrounding micro- and macro-environments [6]. By *in vitro* and *in vivo* mouse models, Lim identified soluble CD44 secreted from GBM cells promotes neuronal degeneration via activating hyper-phosphorylation of tau [15]. This study suggested a direct link between AD and GBM or that GBM might promote neurodegeneration rather than oppose it [15]. Another study posited that neurodegeneration occurs before tumor formation, given that post-mortem GBM samples showed neurodegenerative-associated proteins, such as Aβ and tau, present in the majority of cases but in distant sites and in an age-expected amount and distribution compared to controls [3].

While the two patients reported here suffered from amyloidogenic dementias and only Patient 2 displayed concomitant tauopathy, hyperactivation of mTOR pathway is implicated in both gliomagenesis and amyloid-independent tau aggregation. A recent study employing Tau-PET imaging in adults with tuberous sclerosis complex (TSC), caused by mTOR-activating *TSC1/2* gene mutations, observed significant accumulation of the 3R/4R isoform of phosphorylated tau that is commonly seen in AD. Notably, 5–15% of TSC patients develop subependymal giant cell astrocytoma, an intraventricular CNS WHO grade 1 glioma also thought to arise from constitutive mTOR activation. Although both patients received a molecular diagnosis of GBM based on tumor markers (*IDH*-wildtype, *TERT* promoter mutant), each patient’s tumor also harbored pathogenic *PIK3CA* mutations known to serve as critical activators of mTOR signaling (see Table 1) [16]. The literature also mentioned that TERT seemed to provide neuroprotective qualities in experimental models of neurodegenerative disorders and that manipulations via induction of telomerase in neurons may serve as a protective method against age-related neurodegeneration [17], though this appears to be an early, speculative statement that requires further investigation.

Similar metabolic pathways may be involved in both GBM and dementia, including modifyication of high-density lipoproteins (HDL) via cholesterol esterase transfer protein (CETP). *In vitro*, HDL was found to inhibit GBM cell growth in a nontoxic and dose-dependent fashion, while *CETP* gene functional polymorphism was associated with the incidence of dementia and memory decline [6]. Apolipoprotein E (ApoE) participates in lipid transport and distribution, lipoprotein metabolism, regulation of inflammation and immune activity, maintenance of cytoskeleton stability, and neural tissue function. ApoE (ε4 > ε3 > ε2) has been shown to stimulate the formation of Aβ in the brain and promote the development of AD; *ApoE* is highly expressed in most solid tumors, including GBM, and has been linked to tumor metabolism, immune regulation, and tumor invasion in other cancers [18].

A related protein increased in both AD and GBM is SERPINA3. Increased levels have been associated with brain dysfunction in aging and linked to Aβ development in the brain with increased expression near neuritic plaques in both animal models and AD patients [8]. Regarding gliomas, increased *SERPINA3* expression in patient samples directly correlated with tumor grade and indirectly correlated with patient survival; further preclinical studies revealed SERPINA3 alters immune activity, lymphatic drainage, and cognitive decline [8]. Intersectins are a group of proteins with regulatory function of diverse cellular pathways, such as endocytosis, kinase regulation, and Ras activation. Intersectin 1 is involved in tumorigenesis in several human cancers, including GBM, with recent studies showing intersectin 1-S involved with human glioma cell migration and invasion. In preclinical models, intersectin 1-L appears necessary for learning, memory, and synaptic plasticity by interactions with different proteins participating in neuron formation and development, including the protein Numb, located in dendritic spines and involved in hippocampal neuronal development [19].

A transcriptomic meta-analysis identified a significant number of genes similarly deregulated in AD and GBM, suggesting a direct co-morbidity relationship, specifically 112/198 genes deregulated in the same direction for AD and GBM. Functional analyses revealed eight processes upregulated in AD and GBM were immune-related (innate immunity and interferon alpha/beta signaling), while 14 processes downregulated in both involved synaptic transmission, generation of neurons, oxidative phosphorylation, and neurotransmitter release. Additionally, 11 processes were downregulated in AD and upregulated in GBM related to cell cycle activity and DNA repair [20].

## 5. Limitations

Limitations of our case reports and literature review exist given the nature of presenting just two patients with dual diagnoses of dementia and molecular glioblastoma. The literature presents mostly correlative data that just scratch the surface at our attempt in understanding if any direct, indirect, or mixed relationship is present in this rare comorbid occurrence. While it is not possible to make any absolute generalizations about link between dementia and glioblastoma, there is preliminary evidence from the literature to at least ask the question as to what relationship might exist.

Additional limitations occurred from the molecular testing used on our two patients. For example, analysis was only capable of reliably identifying point mutations and small insertions/deletions (up to 22 bp per the Mayo Clinic Laboratories report) within the targeted regions of IDH1 (codons 113–138 in the Mayo Clinic test and codons 69–138 in the Duke University assay), IDH2 (codons 136–174 in the Mayo Clinic test and codons 126–178 in the Duke University assay), and the TERT promoter region. Alterations outside these regions would thus not be detected, but the most common clinically relevant alterations were included in these assays. False negatives can arise from too few neoplastic cell nuclei and/or too many non-neoplastic cell nuclei. A false negative was of particular concern in the first patient’s lesion, which had low overall cellularity, while both patients’ gliomas were admixed with non-neoplastic brain tissue. These tests are not designed for detecting structural variants, copy number alterations, or large deletions/duplications, so such alterations may be undetected. The tests cannot distinguish between germline or somatic alterations. Rare polymorphisms can also cause false positives or false negatives.

In regard to the chromosomal microarray, the Oncoscan platform has a genome-wide functional resolution of approximately 500 kilobases for non-mosaic deletions and duplications. However, the functional resolution for any given gene locus can vary depending on the size of the copy number alteration, the density of probes targeting the region, the neoplastic cell purity, and the quality of the extracted DNA. The assay cannot rule out balanced chromosome abnormalities or imbalances of chromosomal regions that are not represented by the probes used on the array. Similar to the targeted sequencing assays, chromosomal microarrays in these cases were at risk for false negatives due to low cellularity and neoplastic cell purity.

Additionally, FISH testing relies on targeted evaluation of neoplastic cells to determine the ratio between signals corresponding to the gene locus of interest and signals corresponding to the centromere of the same chromosome. Gains, amplifications, and losses are determined based on the number of neoplastic cells that show an abnormal gene:centromere signal ratio and the numerical value of the gene:centromere ratio. The minimum number of neoplastic cells showing an abnormal ratio and the minimum/maximum ratio required to meet the cutoff can vary significantly among institutions and can greatly influence whether a specimen is considered normal or abnormal for copy number alterations. Testing also requires reliable determination of which nuclei represent neoplastic cells, which, in the case of gliomas with mild atypia, may be difficult. False positives or false negatives can result from focal alterations involving the probe binding sites that do not translate to a copy number alteration involving the entire gene, chromosome arm, or whole chromosome. Inadequate binding of the probes to the DNA in the absence of a true deletion can also result in false positives.

## 6. Conclusions

Although the pathologic natures of these two diseases seem to be at different ends of the spectrum, they may have related elements. In an attempt to comprehend the mixed observations in the medical literature, it appears that specific molecular pathways may prevent and others may promote the development of coexisting AD and GBM with a more complicated interplay of various molecular interactions than perhaps expected at first glance. Further investigations are needed to more fully understand the details of how neurodegenerative dementias and GBM are related—particularly how their co-occurrence is possible and in which particular molecular environments—which could potentially lead to development of improved prognostication, diagnostic tools, and therapeutic interventions for these disease processes.

## Figures and Tables

**Figure 1 brainsci-14-00143-f001:**
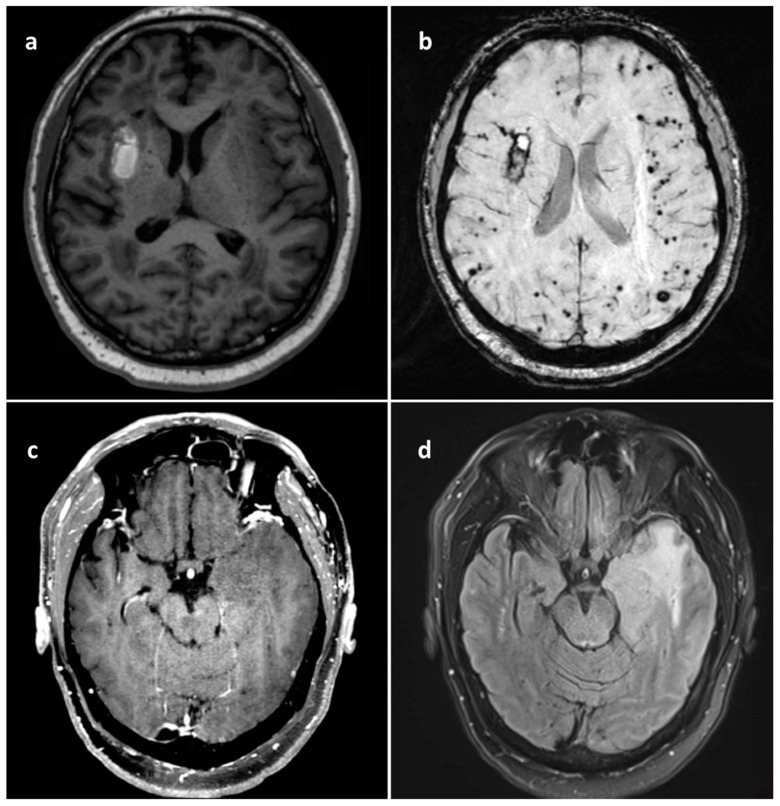
(**a**) Axial MRI T1 without contrast showing right basal ganglia hematoma. (**b**) Axial MRI SWI demonstrating numerous small microvascular hemorrhages predominantly within subcortical white matter of bilateral cerebral hemispheres along with the more peripherally focused right basal ganglia subacute hematoma, overall concerning for cerebral amyloid angiopathy. (**c**) Axial MRI T1 with gadolinium contrast without evidence of any left temporal insular tumor enhancement. (**d**) Axial MRI T2 FLAIR revealing infiltrating left temporal insular tumor.

**Figure 2 brainsci-14-00143-f002:**
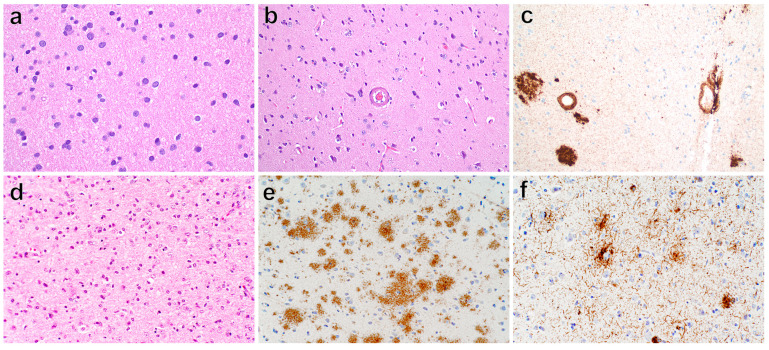
Neuropathologic findings in patient 1 (**a**–**c**) and patient 2 (**d**–**f**). All image scale bars = 50 μm. (**a**) Histology demonstrating an infiltrating glial neoplasm with hyperchromatic, mildly atypical nuclei and occasional nuclear contour irregularities. No mitotic figures, microvascular proliferation, or necrosis (H&E, 400× total magnification). (**b**) Many cortical blood vessels, such as in the center of this image, with thickened walls containing amorphous, eosinophilic material, and separation of the endothelial cells from the intimal surface (H&E, 200× total magnification). (**c**) Immunohistochemical testing for beta amyloid demonstrating eosinophilic material within the vessel walls composed of amyloid, confirming a diagnosis of cerebral amyloid angiopathy. Scattered cortical amyloid plaques (beta-amyloid, clone 6F/3D, Dako, Santa Clara, CA, USA; 200× total magnification). (**d**) Histology demonstrating an infiltrating glioma with moderately increased cellularity, moderate nuclear atypia, and scattered mitotic figures. No microvascular proliferation or necrosis (H&E, 200× total magnification). (**e**) Immunohistochemical testing demonstrating many diffuse amyloid plaques within the neuropil (amyloid precursor protein, clone 4G8, Covance, Princeton, NJ, USA; 200× total magnification). (**f**) Tau immunohistochemistry revealing scattered neuritic plaques as well as rare neuronal inclusions most compatible with pretangles (Tau, clone AT8, Thermo Scientific, Waltham, MA, USA; 200× total magnification).

**Figure 3 brainsci-14-00143-f003:**
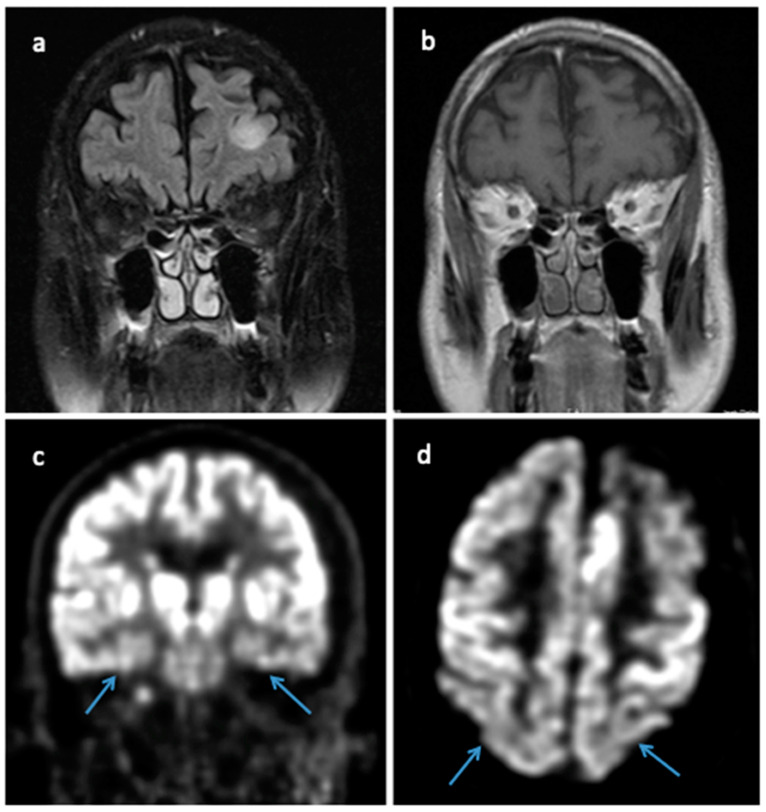
(**a**) Coronal MRI T2 FLAIR showing left frontal hyperintense lesion. (**b**) Coronal MRI T1 with gadolinium contrast without any enhancement of the left frontal lesion. (**c**) Coronal F-18 FDG brain PET scan demonstrating hypometabolism in bilateral temporal lobes (blue arrows). (**d**) Axial F-18 FDG brain PET scan revealing hypometabolism in bilateral parietal lobes (blue arrows).

**Table 1 brainsci-14-00143-t001:** Case comparisons. GBM = glioblastoma, IDH = isocitrate dehydrogenase, TERT = telomerase reverse transcriptase, CAA = cerebral amyloid angiopathy, AD = Alzheimer’s disease, CAD = coronary artery disease, HTN = hypertension, HLD = hyperlipidemia, OSA = obstructive sleep apnea, AA = aortic aneurysm, hypofx = hypofractionated, RT = radiation therapy, TMZ = temozolomide.

	Case 1	Case 2
Sex	Male	Male
Age	68	69
Brain tumor diagnosis	GBM, CNS WHO grade 4	GBM, CNS WHO grade 4
Histopathologic findings	No mitotic figures, microvascular proliferation, or necrosis	Scattered mitotic figures, no microvascular proliferation or necrosis
Molecular findings	*IDH*-wildtype, *TERT* promoter mutation (*C250T*), chromosome 7 gain, likely pathogenic *PIK3CA* mutation (*c.1034A>T*)	*IDH*-wildtype, *TERT* promoter mutation (*C228T*), pathogenic *PIK3CA* (*c.1636C>A*) mutation
Dementia diagnosis	Mixed, CAA & AD	AD
Histopathologic findings	beta amyloid demonstrating eosinophilic material within the vessel walls	abundance of amyloid-β plaques and tau protein
Other comorbidities	CAD, HTN, HLD, OSA, AA without rupture	Benign prostatic hyperplasia, hypercholesterolemia
Social history		
Smoking	No	No
Excessive alcohol	No	No
Illicit drugs	No	No
Brain tumor treatment course	3-week concurrent hypofx RT (total dose: 40 Gy in 15 fractions) with oral TMZ 75 mg/m^2^ daily	3-week concurrent hypofx RT (total dose: 40 Gy in 15 fractions) with oral TMZ 75 mg/m^2^ daily, followed by two 28-day cycles of adjuvant oral TMZ 50 mg/m^2^ daily

## Data Availability

Research data are not publicly available on legal and/or ethical grounds. However, certain questions may be answered by the corresponding author via phone or email contact.

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
