# Peer review of "Coexisting Biopsy-Diagnosed Dementia and Glioblastoma"

_brainsci, 2024, doi:10.3390/brainsci14020143_

Round 1
Reviewer 1 Report
Comments and Suggestions for Authors
Dear authors, I thank you for the honor of being able to review your work. However, the two reported cases do not provide evidence to link dementia and glioblastoma and there are no significant correlations in the reported literature. Please reconsider the objective of the study by reporting two cases of glioblastoma with the relevant comorbidity of dementia and in particular to highlight the compliance of the two patients to glioblastoma therapy. I still ask you to include the section limits of the study and conclusion. In the discussion section, the molecular correlations between dementia and glioblastoma can be highlighted, but specifying that there are no conclusive elements in this sense, while the importance of this comorbidity is interesting from a prognostic point of view.
Author Response
Thank you for taking the time to review our manuscript. We will adjust the paper to more clearly state the limitations of just two case reports and inability to make any absolute generalizations regarding a direct, indirect, mix, or unrelated link between dementia and glioblastoma; however, we have enough evidence from the literature to at least bring this question to the surface and ask what relationships might exist (if any). We will indeed mention more clearly that both patients were compliant with their glioblastoma treatment. We can add limitations statement and conclusion section to the paper as well. We agree that the prognostic importance in addition to any molecular patterns and therefore any potential targetable treatments (whether comorbidity relationship in fact exists or not) are interesting topics to at least mention. We appreciate your efforts in reviewing our work.
Reviewer 2 Report
Comments and Suggestions for Authors
The paper by Fetch-Fayad and coll. is a short case report, on two cases of co-occurrence of glioblastoma in patients diagnosed with dementia. While the description of the cases is straightforward, I hardly see any relationship with the existing literature, cited in the Introduction and in the Discussion. No data are presented that could be used to support a view or the other (e.g., if both disease are linked in some way or not). Since both patients show IDH wild type and mutated TERT promoter, is there any data in the public databases that can indicate some link with dementia? Is it possible to have some indications on the upregulated/downregulated genes in these two patients? This would be very interesting and could be compared with literature data (e.g., those reported by the Authors at lines 245-252). At present, the simple clinical case report seems too preliminary to add informations to the existing knowledge.
MINOR changes:
1. line 53 'seemingly interconnected diseases': the data are scanty. No connection can be postulated based on the present data. Only a note: if no relation exists between GBM and dementia, the percentage of demented patients in the GBM cohorts should be the same as in the non-GBM population. To my knowledge, statistical data supporting any kind of relation between occurrence of GBM and dementia are cloudy, at best.
2. line 132 'repeated MRI': when was it repeated?
Author Response
Thank you for taking the time to review our manuscript. We will adjust the paper to more clearly state the limitations of just two case reports and inability to make any absolute generalizations regarding a direct, indirect, mix, or unrelated link between dementia and glioblastoma; however, we have enough evidence from the literature to at least bring this question to the surface and ask what relationships might exist (if any). We can add a limitations statement and conclusion section to the paper as well. There is no mention of IDH relationship between dementia and glioblastoma. There is one mention of “TERT exhibits neuroprotective properties in experimental models of neurodegenerative disorders suggesting that manipulations that induce telomerase in neurons may protect against age-related neurodegeneration (PMID: 10959037).” We can indeed include this in our paper. Unfortunately, we cannot obtain information on upregulated/downregulated genes in these two patients, though we agree this would be interesting to explore. We can change line 53 to read as ‘potentially or possibly.’ Again, we can be clearer regarding the limited data available to make any kind of sweeping generalizations at this time. The timeline for the ‘repeat MRI’ referred to in line 132 can certainly be added. We appreciate your efforts in reviewing our work.
Reviewer 3 Report
Comments and Suggestions for Authors
In the case report entitled "Coexisting biopsy-diagnosed dementia and glioblastoma”, Fetcko-Fayad et al. describe two cases with co-existing dementia and glioblastoma (GBM), which is evidenced by pathological characterization. This case report appears to be interesting as it may lead to extensive investigations into the potential molecular mechanisms underlying this observation. Some minor concerns listed below should be addressed prior to publication.
Minor concerns:
1. It would be beneficial to add a table showing the demographics of these two cases.
2. Page 3, line 73: “however, molecular studies resulted as isocitrate dehydrogenase (IDH)-wildtype, telomerase reverse transcriptase (TERT) promoter mutation (C250T), and chromosome 7 gain.” It would be great to include a method and material section to describe the molecular studies used for characterizing biopsy tissue. How do you evaluate chromosomal gain? For my understanding, molecular-based analysis can only target to sub-chromosomal regions. Thus, the authors should explain the molecular test methods.
3. Page 3, line 74: “additional testing also revealed a likely pathogenic PIK3CA mutation (c.1034A>T).” and page 4, line 121: “further testing revealed a pathogenic PIK3CA (c.1636C>A) mutation.” If the authors use Sanger sequencing or NGS-based molecular analysis to evaluate the PIK3CA mutation, primer(s) sequence should be listed in this manuscript.
4. Figure 2: Scale bar is missing for IHC images. For Fig.2c and Fig.2f, please includes the information of antibodies used for tissue staining.
5. Page 5, line 154: “Disruption of astrogenic balance appears to be at play in both dementia and GBM, with cell cycle dysregulation leading to the development of malignant astrocytes.” Please link this paragraph to the result section demonstrating the disruption of astrogenic balance.
Author Response
Thank you for taking the time to review our manuscript. We can include a demographics table of the two cases if space allows. Given that the traditional structure for case reports does not typically include a method and material, we are not inclined to add one. However, we can certainly add in the text stating the exact molecular studies used for the patients’ tissue analyses and potentially have our neuropathology colleagues comment on any limitations, which we can later mention in a limitations statement towards the end of the discussion section. We can also have our neuropathology colleague adjust Fig 2 to include the scale bar for the IHC images and antibody staining information. In regards to page 5 line 154, the previous sentence and continuation of this sentence explains the disrupted astrocytic balance with references (“Though GBM induces an immunosuppressive microenvironment and the aging brain is considered more pro-inflammatory, both pathologic states contain reactive astrocytes and immunosuppressant microglial cells [PMID: 34587119]. Disruption of astrogenic balance appears to be at play in both dementia and GBM, with cell cycle dysregulation leading to the development of malignant astrocytes, while astrocytic senescence and cell death without renewal results in neurodegeneration [PMID: 33881971].). Could you please clarify the question if this explanation is not sufficient for a particular reason? We appreciate your efforts in reviewing our work.
Round 2
Reviewer 1 Report
Comments and Suggestions for Authors
Dear authors, thank you for your changes.
Reviewer 2 Report
Comments and Suggestions for Authors
I checked the comments and the Authors’ replies